SOFTWARE

# *paramix*: An R package for parameter discretisation in compartmental models, with application to calculating years of life lost

**Lucy Goodfellow**[1], **Carl A. B. Pearson**[1,2]*, **Simon R. Procter**[1]

1 Department of Infectious Disease Epidemiology and Dynamics, London School of Hygiene and Tropical Medicine, London, United Kingdom, 2 Department of Epidemiology, University of North Carolina, Chapel Hill, North Carolina, United States of America

ॐ These authors contributed equally to this work.
* cap1024@unc.edu

**Data availability statement:** All relevant data are within the manuscript or the associated freely available R package.

## Abstract

Compartmental infectious disease models are used to calculate disease transmission, estimate underlying rates, forecast future burden, and compare benefits across intervention scenarios. These models aggregate individuals into compartments, often stratified by characteristics to represent groups that might be intervention targets or otherwise of particular concern. Ideally, model calculation could occur at the most demanding resolution for the overall analysis, but this may be infeasible due to availability of computational resources or empirical data. Instead, detailed population age structure might be consolidated into broad categories such as children, working-age adults, and seniors. Researchers must then discretise key epidemic parameters, like the infection-fatality ratio, for these lower resolution groups. After estimating outcomes for those crude groups, follow-on analyses, such as calculating years of life lost (YLLs), may need to distribute or weight those low-resolution outcomes back to the high resolution. The specific calculation for these aggregation and disaggregation steps can substantially influence outcomes. To assist researchers with these tasks, we developed *paramix*, an R package which simplifies the transformations between high and low resolution. We demonstrate applying *paramix* to a common discretisation analysis: using age structured models for health economic calculations comparing YLLs. We compare how estimates vary between *paramix* and several alternatives for an archetypal model, including comparison to a high resolution benchmark. We consistently found that *paramix* yielded the most similar estimates to the high-resolution model, for the same computational burden of low-resolution models. In our illustrative analysis, the non-*paramix* methods estimated up to twice as many YLLs averted as the *paramix* approach, which would likely lead to a similarly large impact on incremental cost-effectiveness ratios used in economic evaluations.

**Funding:** The author(s) received no specific funding for this work.

**Competing interests:** We have read the journal's policy and the authors of the manuscript have the following competing interests: In the past 5 years, the authors have received grant funding from the Bill and Melinda Gates Foundation, Wellcome Trust, UKRI, and US CDC.

## Author summary

Researchers use infectious disease models to understand trends in disease spread, including predicting future infections under different interventions. Constraints like data availability and numerical complexity drive researchers to group individuals into broad categories; for example, all working age adults might be represented as a single set of model compartments. Key epidemic parameters can vary widely across such groups. Additionally, model outcomes calculated using these broad categories often need to be disaggregated to a high resolution, for example a precise age at death for calculating years life lost, a key measure when estimating the cost-effectiveness of interventions. To satisfy these needs, we present a software package, *paramix*, which provides tools to move between high and low resolution data. In this paper, we demonstrate the capabilities of *paramix* by comparing various methods of calculating deaths and years of life lost across broad age groups. For an analysis of an archetypal model, we find that *paramix* best matches a high-resolution model, while the alternatives are substantially different.

## Introduction

Mathematical models are essential tools for understanding the transmission of pathogens within populations, for estimating and predicting the associated burden of disease on individuals and health systems, and for helping to inform public health decision-making. One common type of model is the compartmental model, which groups individuals into population compartments reflecting different stages of infection and disease. These compartments are commonly further stratified by other characteristics such as age, location, or risk factors for infection and disease. The resolution for these stratifications is constrained by various considerations, such as data availability and computational resources, and can be broad. In practical terms, such broad groups mean that many individuals who differ meaningfully in epidemiological terms are treated as indistinguishable. As we demonstrate here, these methodological assumptions can substantially impact estimates for decision-making criteria like cost-effectiveness, despite identical empirical inputs.

A notable example of this is age-stratification. Data on model inputs such as population age structure and social contact patterns are often only available in 5-year age brackets and typically use broad open age groups to encompass older ages [1], and higher resolution age brackets require more computational resources, which can be impractical for complex scenario analysis or parameter inference. For simplicity, researchers may elect to align model resolution with interventions under consideration, such as grouping all school-age children when considering the impact of school closures, working-age adults for essential worker programmes, or all elderly individuals for vaccination. Important parameters may vary significantly between individuals within these broad age groups, such as the infection-fatality ratio (IFR), prevalence of co-morbidities and risk factors, or cost of treatment. Modellers must then calculate aggregate values for these key parameters when applying them to discretised age groups. Naive approaches such as using the parameter value at the midpoint or mean age within the group may lead to incorrect results by not accounting for the variation of the parameter within the age group.

Additional issues arise when disaggregating the outcomes of compartmental models to high resolution, such as calculating the distribution of ages at death of individuals within a broad age group. This distribution is useful when calculating the years of life lost (YLLs) in an epidemic, a key measure of premature mortality used in economic evaluations of public

health interventions. YLLs, which are calculated as the remaining life expectancy from the age at which a death occurs [2], often contribute a large proportion of disability-adjusted life years (DALYs) in economic evaluations, and drive the cost-effectiveness of interventions. YLLs are therefore key evidence for investment decisions such as funding routine vaccination programmes. The distribution of ages at death across a broad age group is often assumed to be proportional to the age distribution of the underlying population, but this typically leads to an overestimation of YLLs without also accounting for relative mortality rates across the age group, as deaths may be assigned at younger ages than occur in reality.

To address these issues, we present *paramix*, an R software package which provides functions for modellers to aggregate high resolution data into discrete, correctly weighted model parameters, and disaggregate model outputs into high resolution estimates. The package prioritises practicality, focusing on balancing ease of use with flexibility to support common modelling needs. To demonstrate the impact of different approaches, we compared model outputs using several methods when aggregating IFRs and disaggregating deaths, using an archetypal epidemic model to evaluate vaccination programmes for different underlying populations and pathogens.

## Design and implementation

### Functions

The *paramix* workflow can be summarised as 1) gathering parameters and population distribution, either as functional forms or tabulated data, 2) selecting the model and output resolution, 3) providing 1 and 2 to *alembic()* to create a mixing table for matched aggregation and disaggregation, 4) providing the mixing table to *blend()* to create compartment parameters, 5) simulating models using those parameters, 6) providing model outputs and the mixing table to *distill()* to disaggregate outcomes, and then 7) using the disaggregated outcomes in post-simulation analysis. Fig 1 illustrates this workflow.

The *alembic()* function uses the model and output partitions, e.g. the age groups for the compartments and the age resolution for outcomes, to create a mixing partition. This mixing partition, the union of model age bounds and output age bounds, sets the intervals for calculating weighted parameter integrals (hereafter, weights) and populations (Fig 2). These weights and populations are then combined in different ways for the two stages: according to the model partition for aggregation or according to the output partition for disaggregation.

Let $\mathbb{A} = \{a_i\}$ be the boundaries for the model partition, $\mathbb{B} = \{b_i\}$ the boundaries for the output partition, and $\mathbb{C} = \{c_i\}$ their union, the mixing partition, where all sets are strictly increasing (Fig 2). For brevity, we will denote partitions $[x_i, x_{i+1})$ by their lower bounds, $x_i$ (with $x$ as $a$, $b$, or $c$ as appropriate). The calculations for weights and populations for each mixing partition are then

$$\text{weight}_i = \int_{c_i} \text{parameter}(x)\rho(x)\mathrm{d}x \tag{1}$$

$$\text{population}_i = \int_{c_i} \rho(x)\mathrm{d}x \tag{2}$$

Here, parameter$(x)$ is the user-provided distribution of the parameter of interest, and $\rho(x) > 0$ is the population density per feature of interest, with $\int_{\mathbb{C}} \rho(x)\mathrm{d}x = 1$. Internally, *paramix* handles converting from tabular input into functional forms using base R interpolation functions (*splinefun()* for parameters and *approxfun()* for density), but users can provide a custom interpolation function. In our demonstration, the partition feature $x$ is

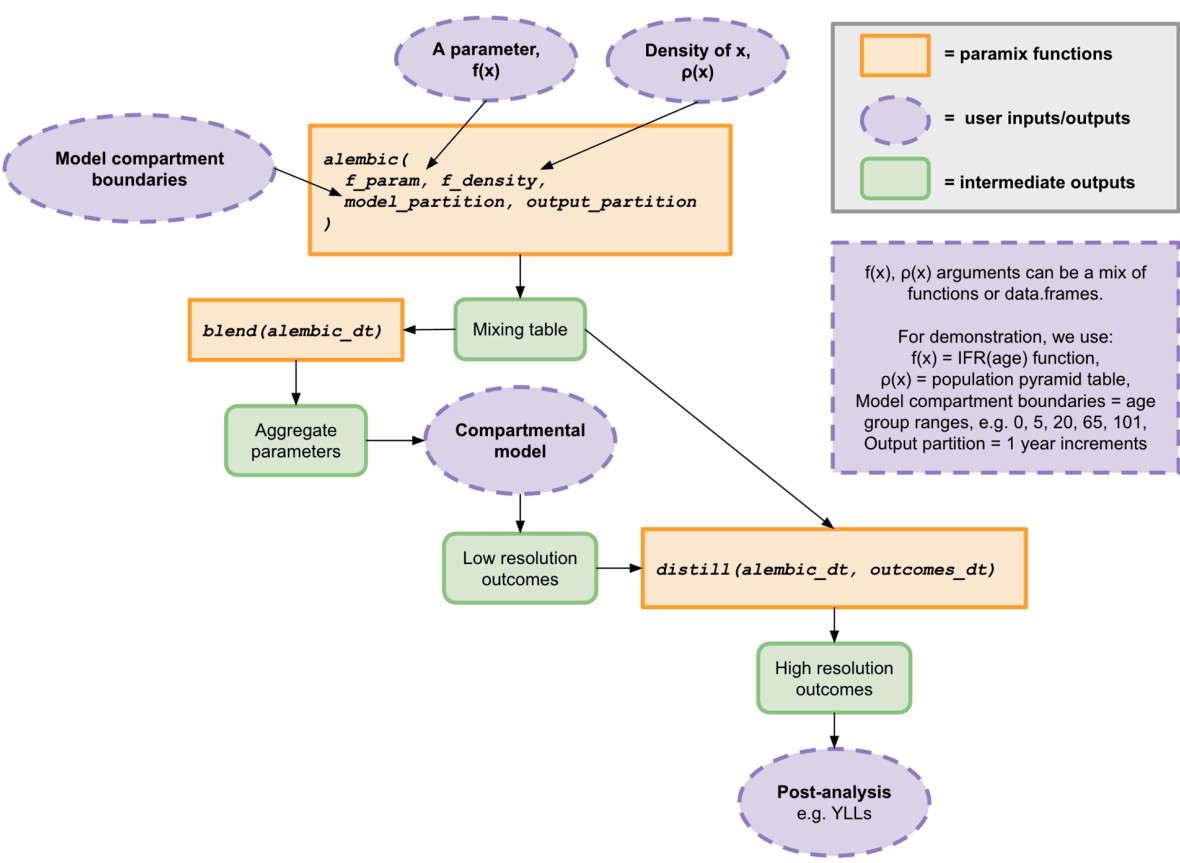

**Fig 1. Functionality of the *paramix* R package and its incorporated functions.**

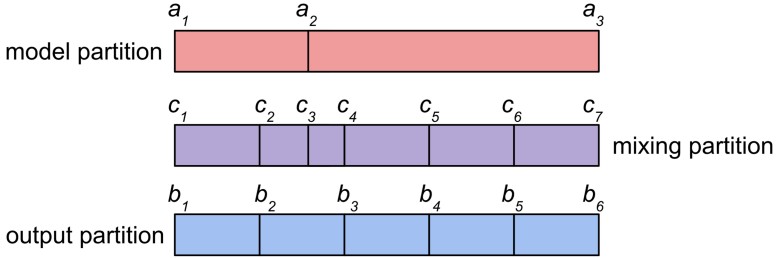

**Fig 2. Example of partitions used in the *paramix* functions, where the model partition (labelled $a_i$, according to their lower bounds) and output partition (labelled $b_i$, according to their lower bounds) may differ, and the mixing partition (labelled $c_i$, according to their lower bounds) is defined as their union.** When the mixing partitions are subsequently used to form parameter discretisation for the model, each $a_i$ includes all overlapping $c_i$, so for example $a_1$ would combine the values from $c_1$ and $c_2$. After simulating outcomes on the model partition, the mixing partition and weights can be used to apportion outcomes, and the overlapping $c_i$ for both the model source and output target partitions are needed. For example, $b_1$ would also take $c_1$ and $c_2$: $c_1$ matches the output partition, but the source partition $a_1$ needs both $c_1$ and $c_2$. In this example, apportioning outcomes to $b_2$ would actually rely on all of the $c_i$, since $b_2$ intersects both $a_1$ and $a_2$ and will draw outcomes from both.

age (so $\rho$ is the density of the population by age), but can be any compartment stratification feature (e.g. risk, if compartments represent discretised behaviour groups, and then $\rho$ would be the population density by risk).

Using the mixing table results, *blend()* then computes parameters for model partition $a_i$ in terms of the corresponding mixing boundary set $\{c_j, c_{j+n}\}$ where $c_j = a_i$ and $c_{j+n+1} = a_{i+1}$

$$\text{parameter}_i = \frac{\sum_{k=j}^{j+n} \text{weight}_k}{\sum_{k=j}^{j+n} \text{population}_k} \tag{3}$$

The *distill()* function distributes model outcomes (for example, estimated hospitalisations or deaths) to the output partition $b_i$ from intersecting model partition(s) in $\mathbb{A}$, i.e. where the lower and/or upper bound of an $a_j$ is within $b_i$. Using Bayes' theorem:

$$\text{P(in partition } b_i|\text{outcome)} = \frac{\text{P(outcome}|\text{in partition } b_i)\,\text{P(in partition } b_i)}{\text{P(outcome)}} \tag{4}$$

The elements on the right hand side can be composed out of the mixture partition weights. Defining the contribution of model partition $a_j$'s outcomes to $b_i$ as $\omega_{ji}$, we can compute $\omega_{ji}$ as 0 (where $a_j$ and $b_i$ do not overlap) or in terms of $c_k = a_j$ and $c_l = a_{j+1}$ (the whole span of the model partition, proportional to P(outcome)), and $c_m = \max(a_j, b_i)$ and $c_n = \min(a_{j+1}, b_{i+1})$ (the span of the intersection, proportional via the same multiplier to the conditioned P(outcome)). Or in terms of set boundaries:

$$\omega_{ji} = \frac{\sum_{c_k \in b_i \cap a_j} \text{weight}_k}{\sum_{c_k \in a_j} \text{weight}_k} \tag{5}$$

Then the model outputs $X_j$ from $a_j$ are transformed into the output partition outputs $Y_i$ in $b_i$ (reiterating that $\omega_{ji}$ is 0 where $a_j$ and $b_i$ do not intersect) by:

$$Y_i = \sum \omega_{ji} X_j \tag{6}$$

Practically, most modelling work will have higher resolution outcome partitions that only subdivide low-resolution model partitions (rather than crossing multiple ones), but *paramix* supports arbitrary intersection of model and output partition bounds.

To see how these relations are implemented in *paramix*, users can refer to the documentation for the *alembic()* function (e.g. via `> ?alembic`) or view the body of the function (e.g. via `> print(alembic)`), which includes inline comments explaining the operations. We have also reproduced those elements in Section 2 of S1 Text.

## Data format

The *paramix* package inputs can be either functions or *data.frame* objects (or any type that extends *data.frame*, such as *data.table* or *tibble* [3,4]), though *paramix* returns *data.table* objects. Model and output partitions are provided as vectors. When users provide the parameter or density 'functions' as tabular data, these are translated into functions via, by default, base R interpolation methods, but this interpolation can also be user-specified.

## Comparison to alternative approximations

To demonstrate the use of *paramix* and compare it to alternative approaches, we consider an archetypal infectious disease dynamical model with age stratification. We aggregate an IFR function to create death-per-infection parameters for the age groups and apply them to that archetypal model. We disaggregate the resulting fatalities based on the underlying population and same IFR function, and then compute YLLs averted by different vaccination programmes. For model age groups, we used four broad groups: pre-school age (0-4), school age (5-19), working age (20-64), and elderly (over 65). We also ran a high-resolution model with 101 age groups, i.e. 0, 1, ..., 100+ year olds, to benchmark the estimates for the broad age groups. We considered vaccination programmes targeting one of the 5-19, 20-64, or 65+ age groups. To ensure comparability of the vaccination programmes, we assumed that a fixed number of doses corresponding to vaccinating 75% of over 65s was available. For each scenario, we allocated those doses to the target age group; for the under 65 targets, this typically yields lower than 75% coverage.

We modelled pathogen dynamics using a Susceptible (S), Exposed (E), Infectious (I), and Recovered (R) epidemic model (SEIR model), with the four age groups as stratifications (Fig 3) [5]. For simplicity, we do not include ageing or births and deaths in the model. Fatalities appear in the R compartment; this is equivalent to living individuals reducing their contact rates as the population size shrinks. We assumed that vaccinations were 50% efficacious with an all-or-nought mechanism, with no loss of immunity in the epidemic time period. We represent vaccination by moving effectively vaccinated individuals to the R compartment prior to the start of simulation. We seeded the epidemic with 0.001% of each age group in the E compartment at the start of the epidemic. For the high-resolution model, both vaccination and seeding are distributed proportionally according to population within the low-resolution bounds.

We considered both flu-like and COVID-like pathogens, with assumed differences in transmissibility, length of infectious and latent periods [6,7], and IFR distributions (Table 1). The force of infection for age group *i* was then calculated as

$$\lambda_i = \beta \times \sum_j \left( \frac{c_{i,j} \times I_j}{S_j + E_j + I_j + R_j} \right)$$

Here, $\beta$ is the transmissibility of the pathogen of interest, $c_{i,j}$ is the daily number of contacts between age groups *i* and *j*, and $I_j$ is the number of infectious individuals in age group *j*.

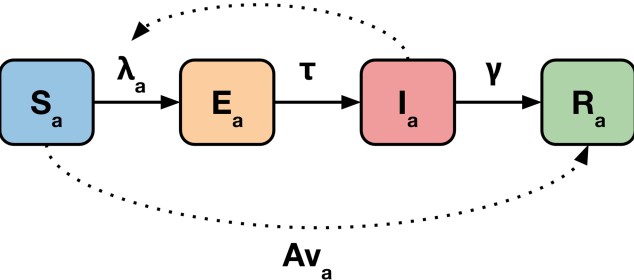

**Fig 3. Age-stratified SEIR model, where $\lambda$ is the force of infection, $\tau$ is the average rate at which exposed individuals become infectious, $\gamma$ is the average rate of recovery, $v$ denotes vaccination coverage, and $A$ vaccine efficacy.** The force of infection is determined by transmissibility, age-specific contact patterns, and the proportion of each age group in the I compartment. Subscript *a* denotes age-specificity.

**Table 1. Epidemiological parameters for the transmission model, for a flu-like and COVID-19-like infection.**

| Pathogen | Flu-like | COVID-19-like |
|---|---|---|
| Transmissibility, $\beta$ | 0.15 | 0.1 |
| Latent period (days), $1/\tau$ | 1 | 3 |
| Infectious period (days), $1/\gamma$ | 2 | 5 |
| Infection-fatality ratio for age $x$ | Proportional to all-cause mortality at age $x$, $m_x$ | $(10^{-3.27+0.0524x})/100$ |

We present results in two underlying populations, using either a rectangular age structure similar to that of many high-income countries (HICs), or a young age structure resembling that of many low- and middle-income countries (LMICs) (Fig 4a). The populations also experienced life expectancies resembling those in HICs and LMICs, respectively, but the same age-specific IFR. These populations were calculated using World Population Project data for the United Kingdom and Afghanistan, respectively [9].

For the IFR, we assumed that the flu-like pathogen IFR follows the all-cause mortality distribution (using mortality data from the HIC setting), while the COVID-like pathogen was associated with a strictly increasing IFR with age [8] (Fig 4b). We scaled the IFRs to produce comparable total mortality for both pathogens. We calculated YLLs as the total sum of remaining life expectancy at death across all fatalities in an epidemic, meaning the results were sensitive to the method of assigning age at death to model fatalities.

We compared three calculation approaches for aggregating IFR when modelling with broad age groups (Table 2). Briefly: the mid-age approach is the simplest and only accounts for the bounds of the age groups; the mean age approach uses the mean age within those bounds based on the population distribution, and the *paramix* approach accounts for age structure across the age group when aggregating IFR. We then compared four calculation approaches for the disaggregation of deaths to calculate YLLs (Table 2). Here, the first approach is again dependent on the bounds of the age groups, the next approach assumes that all deaths occur at the mean age within these bounds, the next assumes that deaths occur proportional to the age distribution, and the *paramix* approach assumes that deaths occur proportional to age and IFR distributions. These approaches represent incremental increases in computational complexity as well as data requirements, but none of them are substantial compared to actual simulation and post-processing demands. The *paramix* approach would be the most complex to implement by hand, but as encapsulated requires the user only invoke 3 commands. When calculating the midpoint of age groups, we assumed that the open-ended 65+ age group ended at age 101.

To review this analysis pipeline, readers may use the follow commands in R to obtain the precise analysis pipeline:

```
path <- "folder/to/copy/to"
srcdir <- system.file("analysis", package = "paramix")
srcfiles <- list.files(srcdir, full.names = TRUE, recursive = FALSE,
 ↪  include.dirs = TRUE)
file.copy(from = srcfiles, to = path, recursive = TRUE)
```

The analysis is orchestrated using the `make` tool, with each step documented in the `Makefile`.

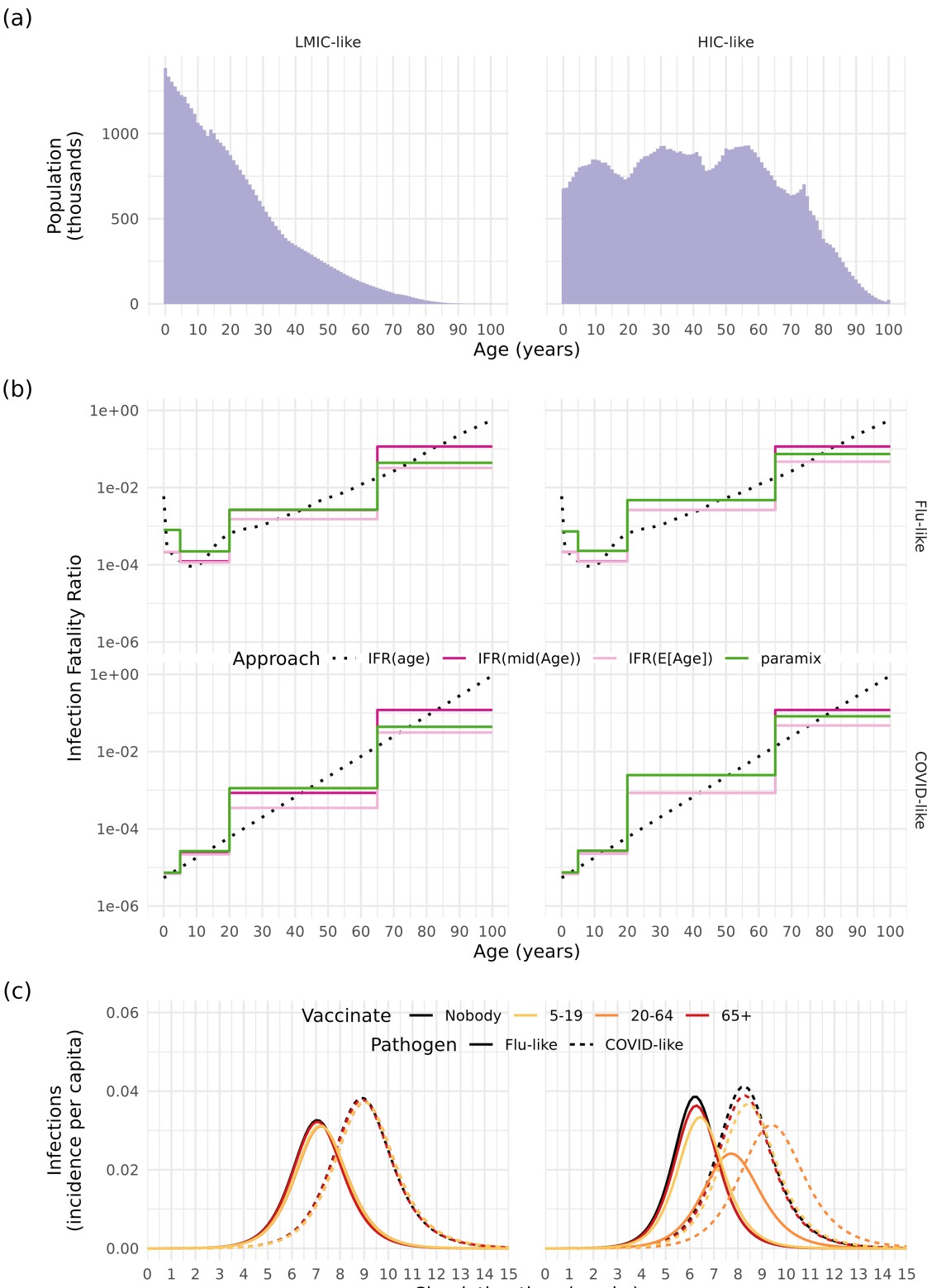

**Fig 4. a. Population age distribution.** b. Age-specific infection fatality ratio, and the results of aggregating into broad age groups using four different methods, for both a flu-like and COVID-19-like infection. c. Incidence of flu-like and COVID-19-like infections, under either no vaccination program or vaccination of specific age groups. Subpanels on the left use the demography of a low- and middle-income country (LMIC); subpanels on the right use the demography of a high-income country (HIC).

**Table 2.** **Aggregation and disaggregation approaches compared in this example.**

| Infection-fatality ratio (IFR) aggregation | | |
|---|---|---|
| **Approach name** | **Calculation approach** | **Formula** |
| IFR(mid(Age)) | IFR of midpoint of age group | $\text{IFR}\left(\frac{(a+b)}{2}\right)$ |
| IFR(E[Age]) | IFR of mean age in age group | $\text{IFR}\left(\int_a^b \rho(\text{age})\,d\text{age}\right)$ |
| E[IFR(Age)] | Mean of IFR across age group | $\int_a^b \rho(\text{age})\text{IFR}(\text{age})\,d\text{age}$ |
| **Age at death disaggregation** | | |
| **Approach name** | **Calculation approach** | **Formula** |
| Uniform | Uniform across age group | $\sim \text{Uniform}[a,b]$ |
| Mean age | All occurring at mean age in age group | $\mathbb{1}\left\{x = \int_a^b \rho(\text{age})\,d\text{age}\right\}$ |
| Prop. to pop. density | Occurring proportional to age distribution | $\frac{\rho(\text{age})}{\int_a^b \rho(\text{age})\,d\text{age}}$ |
| paramix | Occurring proportional to age distribution and mortality | $\frac{\text{IFR}(\text{age})\rho(\text{age})}{\int_a^b \text{IFR}(\text{age})\rho(\text{age})\,d\text{age}}$ |

## Results

### Parameter aggregation

We compared the IFR values for each of the models' broad age groups calculated using each approach; Fig 4b presents the compartmental aggregate values against the true age-specific IFR for flu-like and COVID-like pathogens. The IFR values were identical across populations when using the mid-age IFR (as this approach considers age bounds), but otherwise varied based on the underlying population. The age-specific flu-like IFR increased at very young ages as well as older ages; consequently, the different approaches produced divergent flu-like IFR estimates for the 0-4 model age group. The COVID-like IFR was relatively similar across approaches in the 0-4 and 5-19 age groups, but varied widely in the 20-64 and 65+ age groups.

Incidence per capita varied across underlying populations and between pathogens, as did the comparative effect of vaccinations (Fig 4c). The LMIC-like population experienced a smaller decrease in incidence per capita, due to fewer vaccines used in this example, as the proportion of the population aged over 65 was much smaller than in the HIC-like population (Fig 4a) and we fixed vaccine doses to a matching coverage in the 65+ age group.

In all scenarios, vaccinating those aged over 65 averted the most deaths, but the magnitude of deaths averted and the relative impact of each vaccine programme varied depending on the calculation approach (Fig 5). Using the IFR of the mid-age consistently calculated the most deaths averted, largely due to overemphasis on the eldest in the 65+ age group, while using the IFR of the mean age computed the fewest deaths averted. The relative importance of calculation approaches was the greatest when considering the 65+ vaccination programme: using the mid-age overestimated the number of deaths averted under a COVID-like epidemic in an LMIC-like population compared to the *paramix* approach by 173% when vaccinating those aged over 65, 96% when vaccinating those aged 5-19, and 20% when vaccinating those aged 20-64. Similar findings occurred for flu-like epidemics, and in the HIC-like population.

The estimated number of deaths averted is consistently most similar to the results of the high-resolution model when using the *paramix* approach (Fig 5). However, the high-resolution model run time is around 1000 times greater than the low-resolution models.

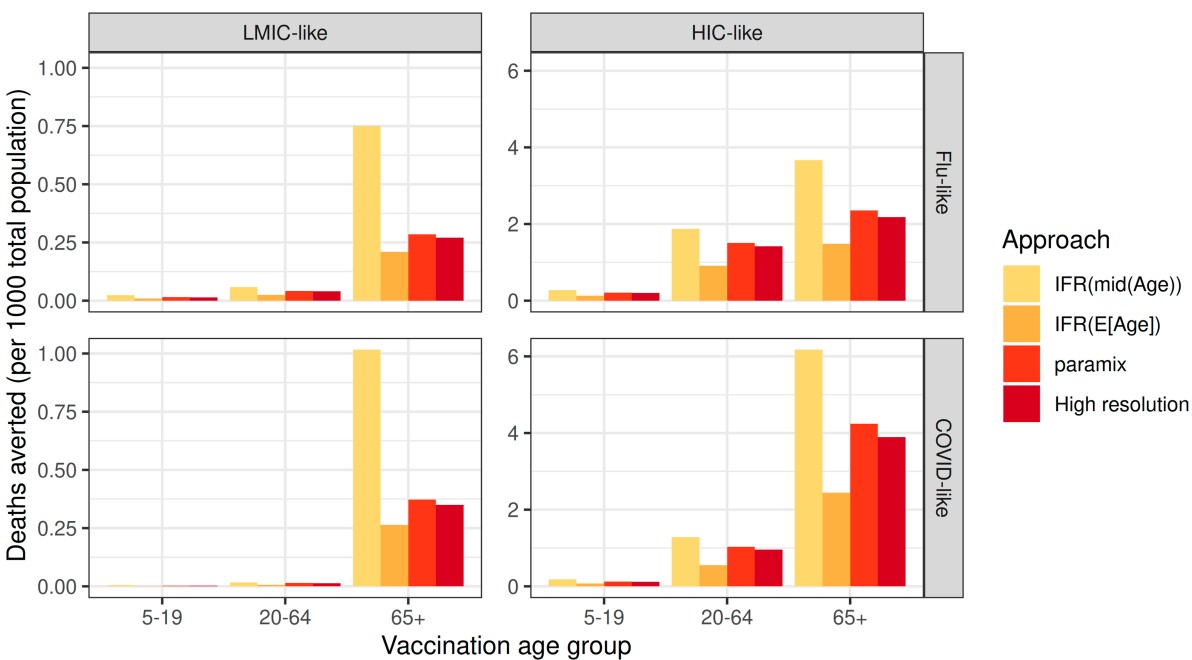

**Fig 5. Estimated deaths averted under each vaccination scenario, for flu-like and COVID-like pathogens in LMIC-like and HIC-like populations.** Shown for three varying approaches of infection-fatality ratio aggregation, including the *paramix* package, and a high-resolution model with no aggregation.

## Outcome disaggregation

The YLLs averted again give qualitatively consistent results for all approaches in all settings, with different quantitative outcomes; however, the preferred intervention is no longer the same for flu-like and COVID-like pathogens (Fig 6). Each YLL computation was based on the same numbers of deaths in each of the four broad model age groups, for comparability (those estimated by the *paramix* approach). This may correspond to research where figures such as the number of deaths in each age group have been provided to researchers who are planning on conducting further economic analysis. Using the mean age or distribution proportional to age only neglects that older individuals are more likely to die if exposed for these IFR trends, which *paramix* accounts for, meaning those approaches assign more deaths to relatively younger individuals compared to *paramix* and thus estimate higher YLLs per death. The effect of this is most extreme when assigning age at death proportional to age distribution only, an approach frequently used by researchers, where for example YLLs averted in an HIC-like population by vaccinating elderly individuals are 99% and 76% higher in COVID- and flu-like epidemics, respectively, compared to the *paramix* approach. The corresponding figures are 65% and 52% in an LMIC-like population. Again, the estimated number of YLLs saved is consistently most similar to the results of the high-resolution model when using the *paramix* approach (Fig 6).

This demonstration has shown that computational approaches for aggregation or disaggregation can drastically change the magnitude of effect of interventions and that using *paramix* most closely resembles the results of a fully disaggregated model. Evaluations which use thresholds to determine if an intervention should be implemented are affected by these changes in magnitude. In some cases, it is also possible that incorrect aggregation and disaggregation of parameters could change the ranking of interventions under consideration,

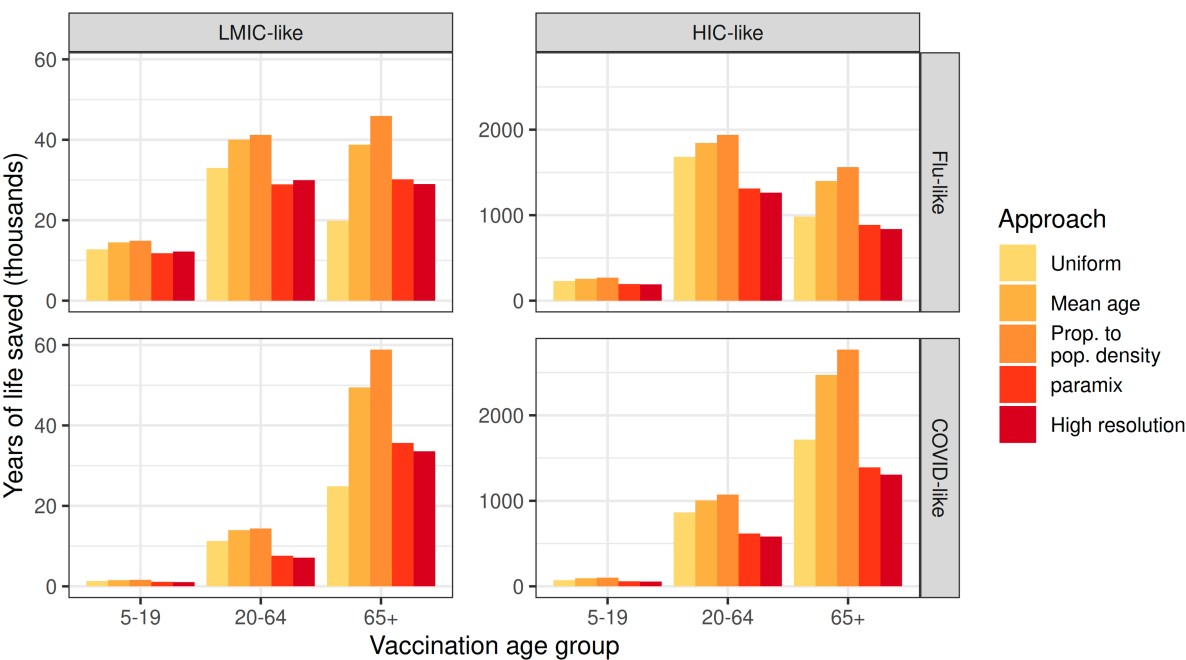

**Fig 6. Estimated years of life saved under each vaccination scenario, for flu-like and COVID-like pathogens in LMIC-like and HIC-like populations.** Shown for four varying methods of age at death disaggregation, including the *paramix* package, and directly from the high-resolution model.

particularly for non-linear parameters. Effective evaluation of public health interventions therefore requires considered and accurate methods of discretisation which take into account the population and parameter densities at hand. The *paramix* package will simplify these processes for modellers and researchers.

### Demonstration limitations

For this demonstrative analysis, we used a relative short time horizon. As is a limitation of any age-compartmentalised model, we would expect *paramix* estimates to diverge from the high-resolution model if there were stronger depletion effects in play: for example, the deaths concentrated amongst oldest population could shift the relative composition of the 65+ age compartment over time and reduce the effective death rate. However, we also generally expect the alternative methods to continue to be more poorly-matched than *paramix*.

We also note that there is a case where the high-resolution and *paramix* estimates suggest a different intervention ranking by a very narrow margin (flu-like epidemic in an LMIC-like population). This is likely due to the exact choice of assigning IFRs for the high-resolution age bands, where a decision still has to be made: for age group $x$ to $x + 1$, we used $\text{IFR}(x)$, when we might have instead used $\int_x^{x+1} \text{IFR}(x)\mathrm{d}x$, $\text{IFR}(x + \frac{1}{2})$, or some other intermediate value. Our interpretation is that these options are within the practical margin of uncertainty for the decision and likely to result in indistinguishable outcomes for this metric. That suggests other policy values may be more appropriate decision factors, for example prioritizing directly protecting vulnerable members of society versus maintaining economic activity by protecting workers (which has its own indirect effects on those vulnerable individuals).

## Availability and future directions

The *paramix* open-source software package is implemented in R and available for download via CRAN (https://doi.org/10.32614/CRAN.package.paramix). Installation instructions, tutorials, and detailed vignettes are available at https://cmmid.github.io/paramix/. Code used in the example detailed in this manuscript is available via Github (https://github.com/cmmid/paramix/tree/main/inst/analysis). This package currently supports only simple interpolations when offered data, but we provide contributor guidelines for anyone that wishes to suggest better defaults and alternatives.

The takeaways of this analysis apply to any age-specific epidemic parameters, or even more broadly, any stratifications of a population. Users can easily compare the effect of different computational approaches for different populations and parameters as we have here by using *paramix*'s builtin summary comparison functions *parameter_summary()* and *distill_summary()* for aggregation and disaggregation, respectively. We have demonstrated that aggregation and disaggregation choices can lead to large and potentially consequential changes in estimated impact of disease interventions, and shown how to use *paramix* to better approximate that impact. We hope that the ease and practicality of *paramix* will help modellers improve their estimates in future work.

## Supporting information

**S1 Text. Section 1. *paramix* functions. Section 2. Documentation for *alembic()*.** (PDF)

## Acknowledgments

We would like to thank Nicholas Davies, Jonathan Dushoff, Thomas Hladish, and Juliet Pulliam for helpful comments on the draft of this manuscript.

## Author contributions

**Conceptualization:** Carl A. B. Pearson, Simon R. Procter.

**Data curation:** Lucy Goodfellow.

**Formal analysis:** Carl A. B. Pearson.

**Funding acquisition:** Carl A. B. Pearson, Simon R. Procter.

**Investigation:** Lucy Goodfellow, Carl A. B. Pearson, Simon R. Procter.

**Methodology:** Carl A. B. Pearson.

**Project administration:** Carl A. B. Pearson.

**Software:** Lucy Goodfellow, Carl A. B. Pearson.

**Supervision:** Carl A. B. Pearson, Simon R. Procter.

**Validation:** Carl A. B. Pearson.

**Visualization:** Lucy Goodfellow, Carl A. B. Pearson.

**Writing – original draft:** Lucy Goodfellow, Carl A. B. Pearson.

**Writing – review & editing:** Lucy Goodfellow, Carl A. B. Pearson, Simon R. Procter.

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
