## [Decision Letter · Decision Letter 0]

11 May 2025

PCOMPBIOL-D-24-02097

paramix : An R package for parameter discretisation in compartmental models, with application to calculating years of life lost

PLOS Computational Biology

Dear Dr. Pearson,

Thank you for submitting your manuscript to PLOS Computational Biology. After careful consideration, we feel that it has merit but does not fully meet PLOS Computational Biology's publication criteria as it currently stands. Therefore, we invite you to submit a revised version of the manuscript that addresses the points raised during the review process.

Please submit your revised manuscript within 30 days Jul 11 2025 11:59PM. If you will need more time than this to complete your revisions, please reply to this message or contact the journal office at ploscompbiol@plos.org. Please include the following items when submitting your revised manuscript:

We look forward to receiving your revised manuscript.

Kind regards,

Nic Vega, Ph.D.

Academic Editor

PLOS Computational Biology

Roger Kouyos

Section Editor

PLOS Computational Biology

**Additional Editor Comments:**

The reviews were positive with regard to the technical advance implemented in the package, but indicated a number of modifications to the manuscript itself that would improve clarity and usability.

**Journal Requirements:**

**Reviewers' comments:**

Reviewer's Responses to Questions

**Comments to the Authors:**

Reviewer #1: The paper is very well-written, easy to read, and is potentially relevant, with practical applications. I only have a few minor comments:

- Within integrals, the $d$ in dx should not be italicized.

- I would bring the definition of mixing table (L62-64, Fig 2) before the introduction of the workflow (L53-60). Now the workflow refers to the mixing table, but the reader can't know what does it mean, as it is introduced only later.

- I would give a unit of measurement for $\rho$.

- Equation (4) is unclear to me: What does "in group [e, f)" mean...? It should be properly defined.

- Equation (5) is unclear to me. "outcome" is never defined, and it is used somewhat inconsistently: on the left-hand-side we have $outcomes_x$, on the right hand side we have $outcomes_x^y$. These should be properly defined.

- I'd add a citation to SEIR, and age-stratified SEIR models.

- "To ensure comparability of the vaccination programmes, we assumed that enough vaccines to vaccinate 75% of over 65s could instead be allocated to different age groups, which typically entails lower coverage but otherwise never approaches 100% coverage." (L102-104) I don't really understand this sentence, could you please elaborate?

- Fig 4b according to the figure caption ("Age-specific remaining life expectancy") is missing from the figure itself. (Thus the remaining ones have wrong letter.) It is also referred to in the text, despite not appearing in the figure.

- It is strange that Figure 4c (actually 4b, see above) is referred before 4a in the text.

- "and approaches 3 and 4 produced similar estimates of deaths averted" (L170) I don't understand this, there are only 3 approaches for aggregation.

Reviewer #2: Goodfellow et al. present an R package to bin parameters that vary as a function of a continuous variable (e.g., mortality rate as a function of age) into a small number of discrete values (e.g., mortality in young, mortality in middle-aged, mortality in elderly). They make a clear argument against using a value at the midpoint of each discrete bin and instead point out that a more statistically justifiable approach would be to take each parameter's expectation over the continuous variable's probability across each bin. While this R package addresses a clear problem, insufficient details are included to fully understand the approach without looking at the source code.

I suggest either going into more formal detail about the maths underlying the methods (more like a methods paper rather than a software paper) or by explicitly working through an example using the R functions in this package (more like a R vignette). For example, equations 1-5 reference bounds a,b,c,d,e,f without explicitly defining; the term “mixing partition” is used without a definition; the parameters to the SEIR model are described only in the figure caption but not explicitly referenced in Table 1 which gives numerical values; etc.

Specific points

• fig 1: purple circles are labeled as “user inputs,” but the circle at the bottom (“post-analysis e.g. YLLs”) seems more like an output. Maybe label purple as “inputs/outputs”.

• line 71: talking about “interpolation” here made me question “what type?” until the later section that says your package uses both linear and spline interpolation. Would be better to give this information when first mention it!

• line 87: the last word on this line (“data.tables”) is all in italics, which implies the name of this class has an “s” on the end. A better phrasing might be “\emph{paramix} returns \emph{data.table} objects”.

• fig 4: The caption needs more detail and should explicitly state that all subpanels on the left follow the LMIC-like distribution and all subpanels on the right follow the HIC-like distribution (also “LMIC” and “HIC” should be defined in the caption, not just the main text). Further, this figure seems to have been changed from an earlier version – some places refer to four panels (e.g., in the figure caption, and in text at bottom of page 5) and in other places three panels (e.g., the figure itself and in text on page 8).

• fig 5 & 6: interpretation would be easier if the four subpanels in these figures were transposed such that the layout matches fig 4 (i.e., LMIC-like is on left, HIC is on right, Flu-like is on top, COVID-like is on bottom).

**Have the authors made all data and (if applicable) computational code underlying the findings in their manuscript fully available?**

Reviewer #1: Yes

Reviewer #2: Yes

PLOS authors have the option to publish the peer review history of their article (what does this mean?). If published, this will include your full peer review and any attached files.

Reviewer #1: No

Reviewer #2: No

**Figure resubmission:**
---

## [Decision Letter · Decision Letter 1]

11 Aug 2025

Dear Dr. Pearson,

We are pleased to inform you that your manuscript 'paramix : An R package for parameter discretisation in compartmental models, with application to calculating years of life lost' has been provisionally accepted for publication in PLOS Computational Biology.

Best regards,

Nic Vega, Ph.D.

Academic Editor

PLOS Computational Biology

Roger Kouyos

Section Editor

PLOS Computational Biology

The reviews indicate that all major concerns have been addressed, with only minor edits suggested for clarity.

Reviewer's Responses to Questions

**Comments to the Authors:**

Reviewer #1: My concerns have been addressed, I suggest the acceptance of the paper.

Reviewer #2: The authors have addressed my concerns with this revision. I noted just one minor error in their expanded method section:

On line 87, double check the inequalities and description. The text says you're checking if an upper/lower bound of a_j is within b_i. I interpret these as referencing the intervals [a_j, a_{j+1}) and [b_i, b_{i+1}). What is $n$ in $a_{j+n}$? Did you mean +1 instead of +n? Regardless, the text doesn't seem to match the inequalities – there is no check if the lower bound of a_j is within the interval [b_i,b_{i+1}). I think you meant to write that you're checking if the two intervals overlap.

**Have the authors made all data and (if applicable) computational code underlying the findings in their manuscript fully available?**

Reviewer #1: Yes

Reviewer #2: Yes

PLOS authors have the option to publish the peer review history of their article (what does this mean?). If published, this will include your full peer review and any attached files.

Reviewer #1: No

Reviewer #2: No

---

## [Editor Report · Acceptance letter]

PCOMPBIOL-D-24-02097R1

*paramix:* An R package for parameter discretisation in compartmental models, with application to calculating years of life lost

Dear Dr Pearson,

I am pleased to inform you that your manuscript has been formally accepted for publication in PLOS Computational Biology. Your manuscript is now with our production department and you will be notified of the publication date in due course.

With kind regards,

Zsofia Freund
